# Effect of Healthy Transitions intervention in improving family planning uptake among adolescents and young women in Western Nepal: A pre-and post-intervention study

**Dipendra Singh Thakuri**[1]*, **Rajan Bhandari**[1], **Sangita Khatri**[1], **Adhish Dhungana**[1], **Roma Balami**[1], **Nana Apenem Hanson-Hall**[2]

**1** Save the Children International, Nepal Country Office, Kathmandu, Nepal, **2** Save the Children US, Fairfield, Connecticut, United States of America

* dipendrathakuri95@gmail.com

## Abstract

### Background

Contraceptive use can prevent unintended pregnancies, early childbearing, and abortion-related deaths. Despite these benefits, the use of modern contraceptives remains low among adolescent girls and young women (AGYW) in Nepal. To address this gap, the Healthy Transitions Project was implemented in Karnali Province, Nepal from February 2019 to September 2021. This study aimed at measuring the effect of Healthy Transitions' intervention on improving knowledge and use of modern family planning methods among AGYW in Nepal.

### Methods

We used a pre- and post-intervention study design to assess the effect of Healthy Transitions project. A quantitative survey was conducted at baseline and after the first cohort of AGYW had completed the intervention (1 year later). The baseline survey was conducted in 2019 with a cohort of 786 married and unmarried AGYW aged 15–24 years. An end line survey was conducted in 2020 with 565 AGYW who were interviewed at baseline. Data analysis was done using STATA version 15.1. The exact McNemar significance probability value was used to decide the significance of difference between baseline and endline.

### Results

The knowledge and uptake of modern family planning methods have increased in the endline compared to the baseline. AGYW recognised 10 out of the ten modern methods at endline, a significant increase from 7 at baseline (p<0.001). Among AGYW, 99% were aware of sources to obtain family planning methods, compared with 92% at baseline (p< 0.001). The proportion of married AGYW using modern contraceptive methods was significantly higher at the endline 33%, than baseline (26%) (p<0.001).

**Data Availability Statement:** All relevant data are within the paper and its Supporting Information files.

**Funding:** This study was supported by Margaret A. Cargill Philanthropies under Healthy Transitions for Nepali Youth Project. The funding body has no contribution and connection to the study design, data collection, analysis, and interpretation of data in writing manuscript.

**Competing interests:** The authors have declared that no competing interests exist.

## Conclusion

Our results show that multilevel demand and supply-side interventions, targeting adolescents and young women, their families, community, and health system helped to improve knowledge and use of modern family planning methods among AGYW. The study suggests that these intervention approaches can be adopted to improve family planning use among adolescents and young women in other similar settings.

## Background

Pregnancy in adolescence and early motherhood remains a global public health concern [1,2]. Adolescent mothers around the world disproportionately experience pregnancy-related deaths and health risks due to early childbearing and repeated pregnancies [3]. An estimated 38 million adolescent women in developing regions intend to avoid pregnancy, among them around 60% of them (23 million) lack access to modern contraceptives resulting in 10.2 million unintended pregnancies, 3.3 million unplanned births, and 5.6 million abortions every year [4].

Family planning (FP) is a key intervention to prevent these adverse health outcomes [5–7]. Voluntary uptake of contraceptives by adolescent girls and young women (AGYW) reduces teenage pregnancy and prevents pregnancy-related health risks including unplanned births, birth complications, and unsafe abortion [6,8]. Past evidence shows that such FP intervention can prevent 90% of abortions, 32% of maternal deaths, 20% of pregnancy-related morbidity globally, and reduce 44% of maternal mortality in low-income countries like Nepal [5,9].

Nepal has made significant progress in FP as evidenced by a reduction in total fertility rate (TFR) from 4.5 in 1996 to 2.3 in 2016 [10]. Similarly, the contraceptive prevalence rate for modern contraception among currently married women increased from 26% in 1996 to 43% in 2006 then remained stagnant through 2006 [11].

Regardless of the substantial progress in FP, inequalities in access to modern FP methods are still evident among adolescents, poor and marginalised women [12]. Nationally, only 15% of married adolescents (15–19 years) use modern contraceptive methods. The unmet need for FP among adolescents continues to remain high (35%), which has negatively influenced adolescent health outcomes due to higher adolescent pregnancy (17%) or childbirth in Nepal [13].

Similarly, while the average fertility rate in Nepal has decreased from 5.1 children per woman (1984) to 2.3 (2016), the adolescent-specific fertility rate has increased from 81 births per 1000 women (2011) to 88 births per 1000 women (2016) [11].

Family planning is a choice for many adolescents and youth. Still, they often experience barriers such as long-distance to health care facilities, negative provider attitudes, and inadequate stock of preferred contraceptives. Nepali youths are reluctant to use modem contraceptives due to misconceptions about long-term fertility risks, fear of adverse reactions, and overall lack of deeper knowledge about FP [5,14,15].

Expanding the coverage and access to effective contraceptive methods among AGYW through effective policies and programs can accelerate progress toward achieving the nation's FP goals and universal access to reproductive health care services by 2030 [11]. Therefore, FP program has been a prioritised and long-standing strategy of the Government of Nepal [12]. This can be seen in the prominence given to FP throughout the country's development plans and strategies (e.g., Nepal Health Sector Strategy (NHSS) 2015–2020, the Population Perspective Plan 2010–2031, and FP Costed Implementation Plan 2015–2020), including commitment for FP 2020 and achieving Sustainable Development Goals (SDGs) by 2030 [16,17].

Moreover, Nepal has implemented FP program and has also promoted adolescent-friendly health services through the adolescent sexual and reproductive health (ASRH) program [12]. Various factors negatively influence the delivery of FP services including lack of information among Adolescents and Youths, lack of trained staff, and various cultural and religious factors [5]. Similarly, the existing programs are limited more to a supply-side approach and focus on an individual level of adolescent and young women. However, the programs have failed to reach husbands, families, and the community, who are key influencers for AGYW's FP decision-making [12].

Thus, to address this gap, Save the Children, in partnership with four local non-governmental organisations (NGOs), designed and implemented the Healthy Transitions for Nepali Youth Project (Healthy Transitions) from February 2019 to September 2021. This project aimed to improve reproductive, maternal, and newborn health (RMNH) and well-being of AGYW ages 15–24 years. The project applied the socio-ecological model [18] and included gender transformative interventions at each level, such as AGYW, their husbands and in-laws, community influencers and health service delivery systems.

Evidence from other countries highlighted the importance of integrated demand and supply-side interventions in improving FP use [19,20] and maternal health services [21,22]. However, there is limited evidence among AGYW in the Nepalese context [23,24]. Hence, this study examines the effect of Healthy Transitions' intervention in improving the uptake of FP services through the analysis of key indicators obtained from the baseline and endline surveys.

## Program intervention

The Healthy Transitions intervention was implemented in 9 Rural/Municipalities of four districts in Karnali Province covering 40 health facilities catchment areas. The project applied a socio-ecological model focusing on demand and supply-side gender transformative interventions at different levels. At the demand side, individual AGYW were engaged through mentor-led curriculum-based small groups sessions where they developed sexual and reproductive health (SRH) knowledge and life skills and challenged harmful gender norms. Male partners and families were engaged through home visits and community-based games about SRH and gender, and communities were engaged through community dialogues and social events. At the supply side, Healthy Transitions strengthened the health system through capacity building of health workers, supervision, and essential equipment and renovation support.

Fig 1 Illustrates the Healthy Transitions demand and supply-side intervention focused on multilevel ranging from individuals, their immediate families, community, and health service delivery systems. The figure also conceptualises the relation between Healthy Transitions intervention at different levels and their intended outcomes for improving knowledge and uptake of FP methods.

## Demand side interventions

**Curriculum-based group sessions among adolescents' girls and young women.**  Curriculum-based, participatory group sessions were carried out among AGYW at the project site. The curriculum-based group sessions were used to improve knowledge and abilities to delay and space pregnancies as well as address deeply rooted social and gender norms among AGYW. The sessions were conducted in 9 Municipalities of project areas among 360 AGYW groups, including 18–22 AGYW in each group. These sessions were conducted separately among married and unmarried AGYW groups. The project mobilised and trained the local mentors for conducting group sessions among AGYW groups. One mentor was assigned to manage three AGYW groups. In addition, the project appointed social mobilisers to provide

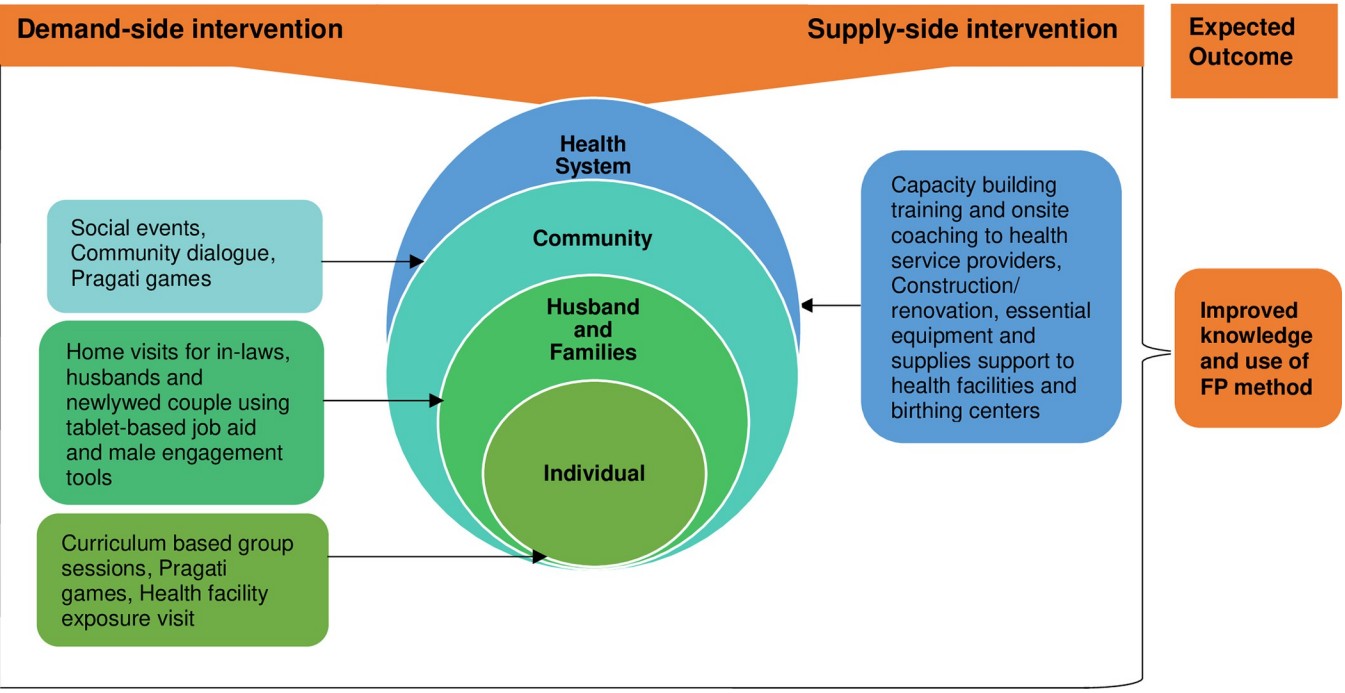

**Fig 1. Schematic representation of multilayer demand and supply-side intervention of Healthy Transitions.**

technical support to mentors to conduct quality group sessions. The Healthy Transition project, under the leadership of the Ministry of Health and Population (MoHP), National Health Education Information and Communication Center (NHEICC), and Department of Health Services (DoHS) developed a standard curriculum called "*Swastha Rupantaran*" (Healthy Transition). The "*Swastha Rupantaran*" curriculum consists of 24 different sessions related to FP and Maternal and Neonatal Health (MNH), such as Menstruation cycle and fertility, Sex and Gender, Sexual and Reproductive Health Rights, Gender-based violence, Sexual Transmitted Infection, HIV and AIDS, basic life skills, financial literacy, and mental health problems in pregnant and post-partum adolescent girls including two sessions that explicitly talk about FP. The two-family planning-related sessions include 1) family planning and contraceptive methods 2) healthy timing and spacing. The FP sessions cover the basics of family planning and its benefits, different FP methods, possible side effects, myths/misconceptions regarding FP use, and healthy timing and spacing. The average duration of 1.5–2 hours session was conducted on a fortnightly basis to complete 24 sessions.

**Health facility exposure visit for adolescents' girls and young women.** The health facility (HF) exposure visit was carried out for AGYW. The exposure visits were conducted to make AGYW aware of the Reproductive Maternal and Neonatal Health (RMNH) including FP services available in their nearest health facilities. The Social mobilisers facilitated exposure visits for all AGYW group members during the project period. During exposure visits, health facility staff provided detailed information about availability of services and the timings of the health facility.

**Home visits to newlywed couples, husbands, and in-laws using tablet-based job aid.** Project developed a tablet-based job aid in close coordination with NHEICC/ DoHS, which consists of six short videos and a user's guide, to be used during a visit with newlywed couples', husbands, and in-laws at their homes. Each of the videos were of an average 6.5 min duration.

Project appointed and trained social mobilisers to conduct home visits in targeted households using tablet-based job aid (videos). The videos were displayed to the target populations at their homes. The videos were used to promote reflection and critical thinking around six different key behaviors and norms related to FP use, gender equity in the household, husband's engagement in newborn and mother's care, mother-in-law's support during husband's absence considering migration, the young wife's role in major household decision making and girls education.

**Pragati game.** Pragati game is an interactive social and behavior change intervention which comprised of nine games adapted from Save the Children's Fertility Awareness for Community Transformation (FACT) project funded by USAID [25]. The game aims to improve knowledge on fertility, promote conversations in communities around fertility, family planning, and social norms that drive birth timing and family size. Mentors played Pragati games on a regular basis, in different settings with the support of social mobilisers. The Pragati game was played separately among different groups such as among AGYW, husbands, and community influencers in the community. The AGYW played Pragati game during group sessions and school settings whereas community influencers and husbands played in the community setting.

### Supply-side interventions

**Training to health workers.** The skilled-based training was provided to enhance the capacity of Health Workers (HWs) to deliver quality FP services. Project supported to train HWs on Long-Acting Reversible Contraceptive (LARC) such as Intra-Uterine Contraceptive Device (IUCD) and Implant, Comprehensive Family Planning Counseling (CoFP), and Adolescent Sexual and Reproductive Health (ASRH) training. The training was provided based on the need assessment in the health facilities (HFs) of the project area and as per the National Health Training Center's (NHTC) training package. From 2019 to 2020, 28 HWs were trained on implant training, 12 Midwives were trained on IUCD training, 31 HWs were trained on CoFP counselling training, and 51 HWs trained on ASRH training.

**Onsite assessment and clinical coaching for Implant and IUCD service provider.** Onsite assessment and clinical coaching were organised to assess the knowledge and skills of the FP service providers and provide onsite coaching support based on the identified gaps. Over the course of the project, NHTC certified coaches provided on-site coaching support to HWs at the health facility level on a regular basis. Also, regular follow-up visits were carried out by the project team to see the progress.

**Essential RMNH and FP equipment's and supplies support to health facilities and birthing centers.** Essential equipment and supplies were provided to different HFs in the project area to improve the quality of FP and MNH services. Based on the need assessment and identified gaps, the equipment support was provided to 40 HFs and five referral hospitals of the project area. The specific equipment supported for family planning services were implant sets, IUCD sets, and FP counseling kits. Apart from these, some other equipment provided were examination table, waiting bed, essential equipment and furniture for health facilities and birthing centers. However, the equipment and supplies support did not cover FP commodities.

## Methods

### Study design and setting

We used pre- and post-intervention study to assess the effect of Healthy Transitions' intervention for improving knowledge and use of modern FP methods among AGYW. The evaluation

of Healthy Transitions was conducted externally using quantitative survey at baseline and after the first cohort of AGYW had completed the intervention (1year later). The baseline survey was conducted in 2019 with a cohort of 786 married and unmarried AGYW (aged 15–24 years) and an endline survey was conducted in 2020 with 565 of the original AGYW sampled at baseline of the first cohort. This study was carried out in the 9 local government areas or municipalities/rural municipalities of four districts in Karnali province, Nepal. The constitution of Nepal 2015 defines local government as rural municipalities and municipalities [26]. Karnali Province is one of the lowest-ranked provinces on the Human Development Index (HDI) [27]. Multidimensional poverty is highly prevalent in this province, where more than half (51%) of its population fall under the poverty line [28]. Over 4 in 10 (44.3%) women aged 15–49 years in the Karnali entered their first marriage before 18 [11]. Of young women aged 15–19 years in Karnali Province, about 15% have begun childbearing [11]. The total estimated population of Karnali (1,570,418) live in (298,174) households. The estimated total population of four Healthy Transitions' districts is 920,826, with 180,261 adolescents and young populations [29]. We selected 9 Municipalities of Karnali considering the high adolescent population, geographical remoteness, and low service uptake rate. Other municipalities were not chosen to avoid duplicating activities that could have been implemented by other organisations working in those areas.

## Study participants

The baseline and endline surveys were conducted among AGYW aged 15–24 years by IMPAQ LLC (the consulting firm charged with designing and conducting the research), and its Nepal-based data collection partner, Solutions.

## Sampling frame, and techniques

The sampling frame for this study includes unmarried and married AGYW aged 15 to 24 years who were invited to participate in the Healthy Transitions voluntarily. We followed a two-stage sampling strategy to randomly select young married and unmarried aged 15–24 years from among a list of project participants. In the first stage, at each site, we randomly selected two mentors using probability-proportional-to-size sampling for a total of 80 mentors. We used the total number of AGYW program participants (married and unmarried) in groups assigned to each mentor as a size measure. This procedure ensured all AGYW program participants had the same chance to be included in the study. In the second stage, 10 AGYW program participants per mentor were randomly selected for the survey to reach a target sample of 800 AGYW respondents. In addition, we included in our list up to four randomly selected extra AGYW participants (per each mentor) as replacements, in case some of the originally selected AGYW participants would not be available for the survey. Out of a list of 1,120 AGYW participants, we were able to identify and contact 956 AGYW and conduct 786 completed surveys in baseline and 565 in endline. Out of 221 AGYW who were not surveyed during the endline, 95.02% migrated for higher secondary education, marriage and moved to India with their husbands for labor work. The remaining 5% were not surveyed because of their inconsistent participation in the project. Furthermore, for analysing family planning use, we have only considered married adolescents and young women, comprising 290 in baseline and 212 in endline.

## Outcome variables

The outcome variables of this study include knowledge and the use of FP methods. We assessed the use of any FP methods and modern FP methods. Similarly, knowledge regarding

FP was assessed among AGYW. FP knowledge was assessed by asking whether they know about different modern contraceptive methods such as Condom (yes/no), Injectables (yes/no), Pill (yes/no), Female sterilization (yes/no), Male sterilization (yes/no), Implants (yes/no), IUCD (yes/no), LAM (yes/no), Emergency contraceptive (yes/no), Standard days (yes/no). Correctly knowing about modern contraceptive methods would mean the respondents had heard about different types of modern contraceptive methods. Similarly, they were assessed on their knowledge about the place to obtain a method of FP (yes/no), knows about the fertile period (yes/no), knows when the woman becomes pregnant before the menstrual period returns (yes/no).

The practice of FP was assessed to understand the use of modern FP methods. Questions related to use of modern FP methods included if the respondent have ever used any modern method or tried in any way to delay or avoid getting pregnant? (yes/no), and the current use of FP methods assessed by whether respondents currently doing something or using any modern FP method to delay or avoid getting pregnant? (yes/no).

## Explanatory variables

Explanatory variables were selected based on the previous studies and a review of previously published literature [11,30]. Variables such as age, marital status, parity, school status, education of respondents, ethnicity, and wealth were categorised for this study. The respondent's age was categorised into 15–19 years and 20–24 years. The marital status of respondents was categorised into two groups: ever married and never married. Similarly, parity was categorised based on birth status—never given birth, 1 birth, and 2 or + birth. Respondent's school status was categorised into two groups: In school and out of school. Also, the education level of respondents was categorised into two groups: Less than 8 grade and grade 8 and above. The ethnicity was categorised into three groups: *Bhramin/Chhetri*, *Dalit*, *Janajati*, and *Thakuri/ Dashnami*. Similarly, the respondent's wealth was categorised into three groups: Lowest, Middle, and Highest. We created a wealth score for each individual using principal component analysis (PCA). We ranked each respondent by their assigned scores and divided them into the wealth categories: women with the lowest, middle, and highest socioeconomic status.

## Data collection tools and techniques

The digitised survey tool was developed based on the literature review [11]. The survey questionnaire consisted of sociodemographic information of the AGYW, knowledge regarding FP, and its use. Data was collected through Survey Solutions, a software allowing it to be administered using mobile phones, allowing for automated skip patterns, and eliminating the need for data entry from paper surveys. The survey tool was pretested to ensure the quality and necessary modification was done in the flow of the questionnaire and language style. The data collection for this study took place twice, before and after the intervention with the same participants. A face to face interview was carried out to collect the data, and each interview lasted for 30-45min. The trained enumerators collected the data before the intervention in February- March 2019 for baseline and after the intervention in March 2020 for endline. The enumerators were trained on the data collection process and research ethics before conducting interviews and the field work, managers and field supervisors worked closely to oversee data quality and to provide the team with technical assistance. Furthermore, several measures were taken to avoid biases and spillover effect in the study. The overall project interventions were divided into two cohorts run sequentially from 2019–2020 and 2020–2021. While community activities occurred for each cohort, the findings for this study were based on a sample of participants from the first cohort only and participants were engaged at the end of their group

activities. Similarly, household members engaged in home visits and partner activities were not repeated across cohorts. Another control measure was to have an external and reputable consulting firm to conduct the data collection and analysis for this study.

## Data analysis

Data analysis was done using STATA version15.1. Collected data were cleaned and cross-checked to ensure consistency. Findings of descriptive analyses were reported with frequency and proportion of knowledge and use of FP methods. The significance of the difference between before and after intervention was tested through McNemar test. The exact McNemar significance probability value was used to decide the significance of difference. All the independent variables were dichotomized before conducting descriptive analysis and running McNemar test.

## Research ethics approval

Ethical approval was obtained from the Ethical Board of Nepal Health Research Council. Before data collection, we obtained permission from the respective municipalities of four study districts. The research assistants obtained informed consent from all participants before the interview and parental verbal consent was obtained for adolescent girls under 18 years of age. The participants were assured of confidentiality and privacy during data collection. Participation of the study respondents was voluntary where participants were provided with the option of terminating the interview at any time. Data collection was done only among participants who agreed to take part in the study.

## Results

Table 1 depicts the distribution of respondents according to sociodemographic characteristics in baseline and endline. The baseline and endline surveys were conducted among 786 and 565 AGYW, respectively. Over 6 in 10 women were between the age of 15–19 years in both the baseline and endline participants. Similarly, most respondents belonged to *Brahmin* and *Chhetri* ethnicity, with *(N = 786, n = 440, 56%)* in the baseline and *(N = 565, n = 320, 58%)* in the endline. There was a similar proportion of AGYW in all four districts in both baseline and endline. Most of the AGYW were in schools, with *(N = 786, n = 504, 64%)* in the baseline and *(N = 565, n = 375, 66%)* in the endline, as show in Table 1.

The attrition analysis revealed that the overall sample at endline had similar characteristics, as measured at baseline, to the sample that was not surveyed at endline. The data presented in Table 2 shows the subgroup composition of AGYW surveyed and not surveyed at endline. However, there are some differences in age and school status. Compared to AGYW in the endline sample, AGYW not surveyed at endline are more likely to be older (39% were 20-24-year-olds at baseline vs 31% of AGYW surveyed at endline), less likely to have been in school at baseline (42% were out of school at baseline vs 34% of AGYW surveyed at endline) (Table 2).

The knowledge on FP and prevalence of FP uses among married AGYW significantly increased after 12 months of intervention. AGYW recognised 10 out of the ten modern methods, a significant increase from 7 at baseline (p<0.001). Similarly, participants were able to mention 11 any FP methods in the endline which was significantly increased from 8 at baseline (p<0.001). Among AGYW, 99% were aware of source to obtain FP methods, compared with 92% at baseline (p<0.001). The change from the proportion of married young women using modern contraceptive methods at baseline (26%) to that at endline (33%) was statistically significant (p<0.001) (Table 3).

**Table 1. Baseline and endline characteristics of Adolescent Girls and Young Women (AGYW).**

| Variables | Categories | N = 786 (Baseline) | N = 565 (Endline) |
|---|---|---|---|
| | | n (%) | n (%) |
| District | Dailekh | 180 (23%) | 131(23%) |
| | Surkhet | 200(25%) | 139 (25%) |
| | Kalikot | 198 (25%) | 150 (27%) |
| | Jajarkot | 208 (27%) | 145 (26%) |
| Age group | 15–19 | 524 (67%) | 389 (69%) |
| | 20–24 | 262 (33%) | 176 (31%) |
| Marriage | Never Married | 496 (63%) | 363 (64%) |
| | Ever Married | 290 (37%) | 202 (36%) |
| Parity | Never given birth | 549(70%) | 400 (71%) |
| | 1 birth | 113 (15%) | 85 (15%) |
| | 2+ births | 124 (15%) | 80 (14%) |
| School status | In school | 504 (64%) | 375 (66%) |
| | Out of school | 282 (36%) | 190 (34%) |
| Education | Less than grade 8 | 251(32%) | 179 (32%) |
| | Grade 8 and above | 535 (68%) | 386 (68%) |
| Ethnicity | Brahmin/Chhetri | 443 (56%) | 327 (58%) |
| | Dalit | 240 (31%) | 171 (30%) |
| | Janajati | 74 (9%) | 50 (9%) |
| | Thakuri/Dashnami | 29 (4%) | 17 (3%) |
| Wealth | Lowest | 272(35%) | 187 (33%) |
| | Middle | 293 (37%) | 224 (40%) |
| | Highest | 221 (28%) | 154 (27%) |

There is variation in the use of modern FP methods by the socio-demographic characteristics of married AGYW (Table 4). Married adolescents and young women in Surkhet district have the highest proportion of using modern FP methods (57%) in endline, increased from (32%) in baseline. But the difference was not statistically significant. The data showed a significant increase in FP use among Dailekh and Kalikot district and both age groups of married young women (15–19 and 20–24 years). Similarly, a ***significant*** increment in FP use was also noticed in the women with other demographic characteristics. After 12 months of intervention, the positive difference in FP use was found among women who are in school (7%, p = 0.001), women from *Janajati* (29%, p<0.001), *Dalit* (18%, p = 0.007) and *Thakuri/Dashnami* ethnicity (18%, p = 0.004), those from the middle (10%, p = 0.001) and lowest wealth quantile (17%, p = 0.022), women with education less than grade eight (17%, p<0.001) and grade eight and above (4%, p = 0.002) and women with no childbirth (6%, p = <0.001) and having one childbirth (6%, p = 0.001). However, the use of modern FP methods was lower among the women with highest wealth quantile, those from Jajarkot district, and who belonged to advantaged ethnic group (Brahmin and Chhetri) in the endline than in the baseline, as shown in Table 4.

## Discussion

This paper has highlighted the results of the Healthy Transitions project, which was designed to respond to the knowledge and behavior gap among AGYW, prevailing social and gender norms related to FP in the community, and critical barriers faced by health facility staff to

**Table 2. Characteristics of AGYW survey respondents, by attrition.**

| Variables | Categories | Surveyed at endline (N = 565) | | Not surveyed at endline (N = 221) | |
|---|---|---|---|---|---|
| | | N | % | N | % |
| **District** | Dailekh | 131 | 23 | 49 | 22 |
| | Jajarkot | 139 | 25 | 69 | 31 |
| | Kalikot | 150 | 26 | 48 | 22 |
| | Surkhet | 145 | 26 | 55 | 25 |
| **Age group** | 15–19 | 389 | 69 | 135 | 61** |
| | 20–24 | 176 | 31 | 86 | 39** |
| **Marriage** | Never married | 363 | 64 | 133 | 60 |
| | Ever married | 202 | 36 | 88 | 40 |
| **Parity** | Never given birth | 400 | 71 | 149 | 67 |
| | 1 birth | 85 | 15 | 28 | 13 |
| | 2+ births | 80 | 14 | 44 | 20 |
| **School Status** | In school | 375 | 66 | 129 | 58** |
| | Out of school | 190 | 34 | 92 | 42** |
| **Education** | Less than grade 8 | 179 | 32 | 72 | 33 |
| | Grade 8 or above | 386 | 68 | 149 | 67 |
| **Ethnicity** | Brahmin/Chhetri | 327 | 58 | 123 | 56 |
| | Dalit | 171 | 30 | 70 | 32 |
| | Janajati | 50 | 9 | 21 | 9 |
| | Thakuri/Dashnami | 17 | 3 | 7 | 3 |
| **Wealth** | Lowest | 187 | 33 | 85 | 39 |
| | Middle | 224 | 40 | 69 | 31 |
| | Highest | 154 | 27 | 67 | 30 |

P-value obtained from the chi-squared test. **p<0.05, ***p<0.01.

provide FP services. In this paper, we examined the key indicators related to knowledge and contraceptives use among AGYW to measure the changes over the course of the project implementation. Based on the results, the knowledge and uptake of modern contraceptives has increased in the endline compared to the baseline. The significant difference of FP use was observed among AGYW of different ages, wealth status, education, school status, parity, and ethnicity. The proportion of changes in FP use was found to be significant among adolescent and young women of both the age groups (15–19 and 20–24 years), those who are in school, women with no childbirth and having one birth, women who had less than eight grade and eighth grades and above education, women with middle and lowest wealth quintile and those who belonged to *Dalit, Janajati and Thakuri/Dashnami* ethnic groups.

The finding indicates that the integrated demand and supply-side interventions at different level (e.g., Individuals, their families, community, and health service delivery system) may have improved knowledge and contraceptive use among AGYW in four districts of Karnali province Nepal.

AGYW in the endline had higher knowledge about modern FP methods and the place to obtain FP services. Women's knowledge about FP is known to influence their use of contraceptive services [31]. The demand-side interventions such as curriculum-based interactive group sessions, home visits, and Pragati games may have helped to increase FP knowledge. The finding of this study is in line with similar studies done in other low and middle-income countries [3,32].

**Table 3. Family planning knowledge and use in baseline and endline.**

| Variables | (Baseline) N = 786 | (Endline) N = 565 | % Change | *P-value* |
|---|---|---|---|---|
| | **N1** | **N2** | **(n = N2-N1)** | |
| **Family planning knowledge** | **N (%)** | **N (%)** | **(%)** | **P-value** |
| Knows place to obtain FP methods | 723 (92%) | 559 (99%) | 7% | <0.001† |
| Knows that a woman can become pregnant before the menstrual period returns | 307(39%) | 356 (63%) | 24% | 0.928 |
| Knows about the fertile period | 291 (37%) | 379 (67%) | 30% | 0.721 |
| **Family planning use (Baseline, n = 290 and Endline, n = 212)** | | | | |
| Use of any methods | 84 (29%) | 81(38%) | 9% | <0.001† |
| Use of modern methods | 55 (26%) | 70 (33%) | 7% | <0.001† |

P-value is Exact McNemar significance probability value.

† Significant at p<0.05.

Similarly, the proportion of married young women using modern contraceptive methods has increased after 12 months of intervention. The result of this study is found to be consistent with a similar study done in Malawi where demand and supply-side interventions led to an increase in Couple Years Protection (CYP) and long-acting reversible contraception (LARC) uptake [19]. The Healthy Transitions intervention such as interactive and participatory group-based sessions, the interactions of husbands and family members after door-to-door home visits, and training and onsite coaching on LARC and ASRH for health workers may have increased the use of FP methods. Furthermore, health facility exposure visits by adolescents provided an opportunity to be familiar with FP services available in the health facility and encouraged them to use FP methods they needed. The uptake of FP methods is negatively influenced by socio-cultural and religious barriers, including lack of FP information, fear of side effects, harmful cultural beliefs, and prevailing deeply rooted gender and social norms [5,33]. The interactive and participatory group sessions may have minimised these barriers through empowering AGYW and improving knowledge and attitude towards FP methods.

Similarly, the household and community level interaction and dialogues among husbands, in-laws, and community influencers using real story-based videos, male engagement tools, and Pragati games may have worked to address the prevailing social and gender norms and build supportive FP attitudes and behavior among husbands and families in the intervention area. The implementation study conducted in India [34] has also highlighted the effectiveness of multi-level community mobilisation interventions to increase modern FP methods.

Furthermore, skilled health care providers, equipment and supplies are critical in family planning programs to meet the supply-side needs for FP provision. Inadequately trained staff could be a barrier to the provision of FP services [35,36]. Therefore, the Healthy Transitions' supply-side intervention that included training and onsite coaching to health care providers including provision of supervision, essential equipment and renovation support may have helped to improve the quality of FP services at the health facility level. A similar supply-side approach was evident to increase the uptake of FP in other countries [34,36].

This study showed variation in the proportion of change in the FP method used by different socio-demographic characteristics of married adolescents and young women. The proportion of changes in FP use was higher among adolescent women aged 15–19 years than those aged 20–24 years. This finding suggested that the project intervention may have provided

**Table 4. Use of modern Family Planning methods in baseline and endline groups of married adolescent girls and young women.**

| Characteristics | Baseline | | Endline | | Change | P-Value |
|---|---|---|---|---|---|---|
| | Total number (N = 290) | Number of FP use (%) | Total number (N = 212) | Number of FP use (%) | | |
| | | N (%) | | N (%) | % | |
| **District** | | | | | | |
| Dailekh | 81 | 22 (27%) | 58 | 21 (36%) | 9% | 0.001† |
| Jajarkot | 66 | 25 (38%) | 53 | 16 (30%) | -8% | 0.101 |
| Kalikot | 69 | 16 (23%) | 50 | 15 (30%) | 7% | 0.001† |
| Surkhet | 74 | 24 (32%) | 51 | 29 (57%) | 24% | 0.371 |
| **Age Group** | | | | | | |
| 15–19 | 52 | 14 (27%) | 48 | 18 (38%) | 11% | 0.001† |
| 20–24 | 238 | 73 (31%) | 164 | 63 (38%) | 8% | 0.003† |
| **Parity** | | | | | | |
| Never given birth | 53 | 6 (11%) | 29 | 5 (17%) | 6% | <0.001† |
| 1 birth | 113 | 34 (30%) | 83 | 30 (36%) | 6% | 0.001† |
| 2+ births | 124 | 47 (38%) | 100 | 46 (46%) | 8% | 0.148 |
| **School status** | | | | | | |
| In school | 250 | 73 (29%) | 182 | 66 (36%) | 7% | 0.001† |
| Out of school | 40 | 14 (35%) | 30 | 15 (50%) | 15% | 0.192 |
| **Education** | | | | | | |
| Less than grade 8 | 94 | 24 (26%) | 53 | 23 (43%) | 17% | 0.005† |
| Grade 8 and above | 196 | 63 (32%) | 159 | 58 (36%) | 4% | 0.002† |
| **Ethnicity** | | | | | | |
| Brahmin/Chhettri | 142 | 50 (35%) | 111 | 36 (32%) | -3% | 0.001† |
| Dalit | 112 | 29 (26%) | 80 | 35 (44%) | 18% | 0.007† |
| Janajati | 24 | 5 (21%) | 14 | 7 (50%) | 29% | 0.013† |
| Thakuri/Dashnami | 12 | 3 (25%) | 7 | 3 (43%) | 18% | 0.004† |
| **Wealth** | | | | | | |
| Lowest | 104 | 26 (25%) | 31 | 13 (42%) | 17% | 0.002† |
| Middle | 103 | 28 (27%) | 57 | 21 (37%) | 10% | 0.001† |
| Highest | 83 | 33 (40%) | 124 | 47 (38%) | -2% | 0.033† |

P-value is Exact McNemar significance probability value.

† Significant at p<0.05.

significant contribution and benefits among the married adolescent women and their husbands to influence the use of FP methods. Additionally, the young women with no childbirth or having one child had increased their use of FP methods compared to the women with two or more children. This could be because the women with no children or having one child may wanted to delay their pregnancy or keep spacing for next child and hence were encouraged to use contraceptives to delay or space their childbirth after being exposed to the Healthy Transitions interventions.

The findings are in line with the trends of FP utilisation as reported in the most recent Nepal Demographic Health Survey (NDHS) 2016 and other previous study from abroad [11]. It is to be noted that the Healthy Transitions intervention were supportive of the women having the lowest and middle wealth status, and belonged to disadvantaged ethnic group (Dalit, and Janajati). Previous studies conducted in Nepal [31,37] and other countries [38,39] have shown that these groups of women are less likely to use FP methods due to the lack of information on FP, prevailing social and gender norms, and poor decision-making power to use health

services. Thus, home visits by social mobilisers using tablet-based videos, male engagement tools, integration of Pragati games in the group sessions, and community dialogues, which are designed to reach the AGYW with no education and from the poor and marginalised communities [26] were probably supportive of making AGYW aware and empowering through meaningful engagement and participation in the intervention components.

While most of the districts had positive changes in contraceptive use at endline than baseline, only Dailekh and Kalikot had a statistically significant increase in modern contraceptive use. A plausible explanation could be that both the districts are comparatively poor in terms of geographical accessibility, road facility, and access to health facilities with more choices of contraceptives. So, Healthy Transition's multilevel interventions could have been supportive to address those gaps.

These positive findings should be interpreted with some limitations in mind. First, regarding the limitation of intervention design, RCT or a quasi-experimental study with a comparison group was not feasible given implementation and budgetary constraints which may affect the interpretation of the results. Second, while we have data from the young women who made life transitions over the past year, given that the majority of the unmarried AGYW in our sample are quite young, data collected one year later does not capture the long-term impact the program may have on age at marriage, first childbirth, use of family planning, and birth spacing. Third, some of the important covariates such as desire for additional children, sex of living children, women decision-making/autonomy power, currently living with husband or not, and exposure to media that previous studies found important predictors of contraceptive practices were not included in this study [13,39]. Fourth, the self-selection of AGYW who signed up to participate in Healthy Transitions. These proactive populations are generally more educated than the average population. Lastly, we rely on self-reported data, which may be biased toward socially desirable answers. Also, the attrition analysis has found differences in some characteristics i.e., age and school status, of AGYW sample not surveyed at endline. Thus, these differences should be kept in mind while interpreting overall findings. Despite the limitations, our study provides important information about the contribution of multilevel demand and supply-side interventions to address the FP needs of most underserved groups, such as AGYW.

## Conclusion

This study suggests that integrated demand and supply-side interventions targeting multilevel target beneficiaries such as adolescent and young women, their immediate family including husbands and in-laws, community members, and health systems may positively improve knowledge and use of FP methods among AGYW. By adopting these intervention strategies, the national family planning program could help to improve the use of contraceptive methods among AGYW in other similar resource-poor and rural settings. Healthy Transitions has seen the start of sustainability as some adolescent girl and young women groups have converted to community-based mother's groups and registered with their local government. Several local governments have also allocated budget for the Healthy Transitions activities to continue within local activities beyond the life of the project. Future research on the cost effectiveness and efficiency of scaling and institutionalising Healthy Transitions could serve as markers for the sustainability of the gains from Healthy Transitions.

## Supporting information

**S1 Data.**
(PDF)

## Acknowledgments

The authors would like to thank the Ministry of Social Development, Karnali Province as well as the four local partners: Social Awareness Center Nepal, Surkhet; Karnali Integrated Rural Development and Research Center, Kalikot; Everest Club, Dailekh; Panch Tara Yuwa Samrakshak Manch, Jajarkot who implemented the intervention in Karnali Province, Nepal. In addition, the authors are grateful to the Save the Children International field staff who led day-to-day implementation in each district and provided technical backstopping to the partners, Gita Shah, Chandra Singh Sejuwal, Khadak Karki, Adweeti Nepal, and Kripa Shrestha. The authors acknowledge the valuable contribution of IMPAQ LLC and its Nepal-based data collection partner, Solutions, for conducting the evaluations. Most importantly, the authors are grateful to the adolescent girls and young women, their partners, families, and communities for engaging with the interventions and sharing their insights and learning with us through the evaluation.

## Author Contributions

**Conceptualization:** Dipendra Singh Thakuri.

**Data curation:** Dipendra Singh Thakuri, Rajan Bhandari.

**Formal analysis:** Dipendra Singh Thakuri, Rajan Bhandari.

**Methodology:** Dipendra Singh Thakuri, Rajan Bhandari.

**Software:** Rajan Bhandari.

**Writing – original draft:** Dipendra Singh Thakuri.

**Writing – review & editing:** Dipendra Singh Thakuri, Rajan Bhandari, Sangita Khatri, Adhish Dhungana, Roma Balami, Nana Apenem Hanson-Hall.

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
