## [Decision Letter · Decision Letter 0]

15 Jul 2022

PONE-D-22-10311Improving family planning uptake among adolescents and young women in Western Nepal: Learning from the implementation of HTNYP ProjectPLOS ONE

Dear Mr. Dipendra S Thakuri,

Thank you for submitting your manuscript to PLOS ONE. After careful consideration, we feel that it has merit but does not fully meet PLOS ONE’s publication criteria as it currently stands. Therefore, we invite you to submit a revised version of the manuscript that addresses the points raised during the review process.

We look forward to receiving your revised manuscript.

Kind regards,

Tsegaye Lolaso Lenjebo, MPH

Academic Editor

PLOS ONE

Journal Requirements:

Reviewers' comments:

Reviewer's Responses to Questions

**Comments to the Author**

1. Is the manuscript technically sound, and do the data support the conclusions?

Reviewer #1: Partly

Reviewer #2: Partly

2. Has the statistical analysis been performed appropriately and rigorously? 

Reviewer #1: No

Reviewer #2: Yes

3. Have the authors made all data underlying the findings in their manuscript fully available?

Reviewer #1: Yes

Reviewer #2: Yes

4. Is the manuscript presented in an intelligible fashion and written in standard English?

Reviewer #1: Yes

Reviewer #2: Yes

5. Review Comments to the Author

Reviewer #1: This manuscript aimed at measuring the effect of the Healthy Transitional for Nepali Youth Project (HTYP) in increasing FP knowledge and use among adolescent girls and young women in Karnali Province of Nepal. The authors conducted a pre and post-intervention survey with adolescent girls and young women (AGYW) aged 15-24 (786 in the baseline and 565 in the endline) and used descriptive analysis to assess the impact of the intervention. The study concluded that the intervention is effective in increasing knowledge and use of contraception among AGYW and suggested scaling up the intervention. Though this manuscript addresses one important under-research topic, I have several comments on data, method of analysis, and conclusions. Addressing these comments would strengthen this paper.

Abstract: Some information on sample size and measurement is not consistent and clear. For example, the sample size in the endline survey is reported 565 in the abstract but in the sampling frame section (page 10) it is reported 558. Similarly, in the abstract, the knowledge of FP method is mentioned but in the method section (under outcome variable, page 10) – the ‘correct knowledge of the FP method’ is mentioned. The author never defines ‘correctly knowledge of FP methods’ in the manuscript. The abstract needs to be revised based on what revisions the authors would like to make based on my comments on the method section below.

Background: Page 3, third para – the author said that the prevalence rate of modern FP methods among currently married women in Nepal noticeably increased from 26% in 1996 to 43% in 2016 which is not really true. In fact, the contraceptive prevalence rate among married women has been stagnant at 43% since 2006 in Nepal.

Page 4, last para – Authors claim that there is a limited body of knowledge about the impact of integrated demand and supply-side intervention in improving FP use, maternal health etc in Nepal – this is not true. These pieces of evidence are available for women of reproductive ages but not for adolescent and young women.

Page 5 – Program Intervention – author said that the project is implemented in 9 local government areas of four districts – what does ‘local government areas’ mean? Is it a rural municipality or urban municipality or ward? Since the author recommended for scale up the intervention in their conclusions – it would have added value if the authors can provide the cost of intervention of this project.

Page 6 – demand-side intervention – It is not clear whether the ‘Swstha Rupantaran’ sessions was organized separately for male or female? Was it a combination for both sexes? What about married and unmarried – being one the culturally sensitive topic it is important to describe how it was organized and what was reactions of participants to the sessions.

Page 7 – Healthy visits to newlywed couples – Paper is not clearer on how the video was displayed, how long was the video?

Page 7 – Pragati game – manuscript would be improved if the information on when the game was played, how many times it was played, whether it was played together with AGYW, husband, community influencers etc or organized separately to each group?

Page 8 – Support to a health facility – not clear in the manuscript whether or not FP commodities were also supplied to the health facility as part of the supply-side intervention. If yes, what FP commodities were supplied.

Methods:

Page 9, Sampling frame: The sample size in the endline is inconsistent with the abstract (565 VS 558). I am also concerned about the effective sample size for the analysis, particularly for the contraceptive (212 young married only) – the authors may want to reflect this on the limitation of the study and recommend a larger study before suggesting to scale-up of the intervention.

It appeared that end line survey was conducted when there was an active intervention, this may bias the overall reporting. Adding an explanation about the approach the authors have taken to control this in the data collection and in analysis would improve the manuscript.

Outcome variables – How authors have defined ‘correctly know about modern contraceptive method’? Also, the authors assessed knowledge about the place to obtain a method of FP (yes/no) – it is not clear how they measured it? For example, one participant may know the place to obtain ‘Oral Pills’ and ‘Male Condom’ but not an IUD, so how is this participant categorized as a dichotomous variable?

Explanatorily variables: Few important explanatory variables such as desire for additional children, sex of living children, women decision-making/autonomy power, currently living with husband or not, and exposure to media are not included in the analysis. Including these important variables in the analysis would improve the paper significantly. Furthermore – it is not clear in the paper how the authors have created a ‘wealth’ variable.

Data collection: One of the major limitations of this study is the high loss to endline interviews and no mention of attrition analysis. What were the main reasons for the loss to follow up (in endline interviews)? Adding analysis about lost to follow-up participants (from the baseline data) would improve the paper. It is also not clear whether or not these were repeated measurements (same respondents interviewed twice??). Was the data collection paper-based? How long was the interviews?

Data analysis: Descriptive analysis is used – if this is a repeated measurement why other statistical methods such as the general linear model for repeated measurement was not used? Justification/rationale of selection of variables, and use of higher-level statistical methods to control the effect of explanatory variables on the outcomes variables would really improve the results and conclusions of this manuscript. This may enhance the author’s confidence in reporting the impacts of the intervention on increasing knowledge and use of contraceptives.

Results: Categorization in ethnicity in Table 1 and Table 3 do not match. Making consistency in these two Tables would improve the paper.

Table 3 – There is a reduction in FP use from baseline to endline among a few groups, for example, in Jajarkot district (38% VS 30%) and among Brahmin/Chhetri ethnicity (35% VS 32%) – I would like to know the likely reason for this? Is it because of data quality?

Discussion and conclusions: The discussion and conclusion section needs to be revised based on how the authors would address my comments on data analysis. Based on the current analysis, I am not fully convinced by the authors’ claim of their intervention has improved knowledge and use of modern contraceptives. The improvement observed between baseline and end-line could be a natural change over time. Therefore, I suggest either improving the data analysis or recognizing these important limitations and revising the discussion and conclusion section of the manuscript accordingly.

Reviewer #2: The work is well written with good english.

My comments are attached with track change and this paper should be published if only the authors amend those important comments forwarded in the result and conclusion sections! Otherwise, the discrepancies and interpretation of some of the data highlighted in the result sections will affect the acceptance of this paper for publication in this journal.

6. PLOS authors have the option to publish the peer review history of their article (what does this mean?). If published, this will include your full peer review and any attached files.

Reviewer #1: No

Reviewer #2: No

---

## [Author Response · Author response to Decision Letter 0]

2 Sep 2022

Aug 31, 2022

Dear Editors-in-Chief

PLOS ONE 

Thank you very much for your email with the decision dated July 15, 2022. We found that reviewers' and editor feedbacks were insightful. We have addressed both reviewers’ comments point by point. We have uploaded the revised manuscript showing track changes so that you can see all revisions and modifications we have made. We believe that our revisions will satisfy you and both reviewers. We have included a clean copy and track change copy of the revised manuscript. 

Thank you for considering this manuscript for publication.

Sincerely,

Dipendra Singh Thakuri

Reviewer #1: 

♣ Abstract: Some information on sample size and measurement is not consistent and clear. For example, the sample size in the endline survey is reported 565 in the abstract but in the sampling frame section (page 10) it is reported 558. Similarly, in the abstract, the knowledge of FP method is mentioned but in the method section (under outcome variable, page 10) – the ‘correct knowledge of the FP method’ is mentioned. The author never defines ‘correctly knowledge of FP methods’ in the manuscript. The abstract needs to be revised based on what revisions the authors would like to make based on my comments on the method section below.

Authors response: Thank you for your comments. The actual sample size in the endline was 565. The 558 was a typo. We have corrected it in the revised manuscript (page 10). In terms of correct knowledge of the FP methods, we have revised this (page 11). Correctly knowing about modern contraceptive methods would mean the respondents had heard about different types of modern contraceptive methods such as Condom (yes/no), Injectables (yes/no), Pill (yes/no), Female sterilization (yes/no), Male sterilization (yes/no), Implants (yes/no), IUCD (yes/no), LAM (yes/no), Emergency contraceptive (yes/no), Standard days Method (yes/no).

♣ Background: Page 3, third para – the author said that the prevalence rate of modern FP methods among currently married women in Nepal noticeably increased from 26% in 1996 to 43% in 2016 which is not really true. In fact, the contraceptive prevalence rate among married women has been stagnant at 43% since 2006 in Nepal.

Authors response: Thank you for your comment. We agree with you that the contraceptive prevalence rate has been stagnant since 2006. We have rephrased the statement in the introduction section as suggested (Page 3).

♣ Page 4, last para – Authors claim that there is a limited body of knowledge about the impact of integrated demand and supply-side intervention in improving FP use, maternal health etc in Nepal – this is not true. These pieces of evidence are available for women of reproductive ages but not for adolescent and young women.

Authors response: Thank you so much for your important comment. Agreeing with your comments, we have revised the introduction section (page 5) to reflect this and incorporated the relevant papers as references

♣ Page 5 – Program Intervention – author said that the project is implemented in 9 local government areas of four districts – what does ‘local government areas’ mean? Is it a rural municipality or urban municipality or ward? Since the author recommended for scale up the intervention in their conclusions – it would have added value if the authors can provide the cost of intervention of this project.

Authors response: Thank you for your insightful comment. We have added details about local government in the revised manuscript (page 9). 

In terms of the cost of the intervention, we recognise the value of costing studies (Rosen et al., 2019) and agree that it would add value to the case for scaling up the intervention. However, since a costing study was not conducted for Healthy Transitions, we could not provide such costing information. Future cost analyses would enhance discussions on scalability of the intervention, and we have added this as a recommendation in the conclusion section (page 21) of the revised manuscript. 

♣ Page 6 – demand-side intervention – It is not clear whether the ‘Swstha Rupantaran’ sessions was organized separately for male or female? Was it a combination for both sexes? What about married and unmarried – being one the culturally sensitive topic it is important to describe how it was organized and what was reactions of participants to the sessions.

Authors response: Thank you so much for your comment. Our “Swastha Rupantaran” sessions were organised only for adolescent girls and young women groups and these sessions were carried out separately for married and unmarried groups. We have added details in the manuscript (page 6). 

♣ Page 7 – Healthy visits to newlywed couples – Paper is not clearer on how the video was displayed, how long was the video?

Authors response: Thank you for your query about home visits. We have added the details in the description of the interventions section (page 7 in the revised manuscript. “These videos were used during home visits and displayed among newlywed couples’, husbands, and in-laws at their homes”. There were six videos each averaging 6.5 min duration. 

♣ Page 7 – Pragati game – manuscript would be improved if the information on when the game was played, how many times it was played, whether it was played together with AGYW, husband, community influencers etc or organized separately to each group?

Authors response: Thank you so much for your valuable suggestion. We have added a description in the revised manuscript (page 7-8). “The Pragati game was played among different target beneficiaries and stakeholders such as AGYW in group sessions and school settings, husbands, and community influencers in the community setting. The Pragati game was played separately among different groups.” 

♣ Page 8 – Support to a health facility – not clear in the manuscript whether or not FP commodities were also supplied to the health facility as part of the supply-side intervention. If yes, what FP commodities were supplied.

Authors response: Thank you so much for your important comment. We did not supply health facilities with FP commodities. Instead, we provided essential FP-related equipment such as examination tables, LARC insertion and removal sets, autoclave sets, surgical drum and FP counseling kits etc. We have clarified this in the revised manuscript (page 8-9).

♣ Methods:

♣ Page 9, Sampling frame: The sample size in the endline is inconsistent with the abstract (565 VS 558).

Authors response: Thank you for noting this discrepancy. The 558 was a typo, which we have corrected.

I am also concerned about the effective sample size for the analysis, particularly for the contraceptive (212 young married only) – the authors may want to reflect this on the limitation of the study and recommend a larger study before suggesting to scale-up of the intervention

Authors response: Thank you so much for your insightful suggestion. The overall sample size (786) was calculated based on the population of program participants and was proportionally representative of the various demographics used in the study. Given that FP use was only among married adolescent girls and young women, however, we only used that sub-sample (290 at baseline and 212 at endline) for the purposes of this analysis. The details have been included in the revised manuscript (page 10). 

♣ It appeared that end line survey was conducted when there was an active intervention, this may bias the overall reporting. Adding an explanation about the approach the authors have taken to control this in the data collection and in analysis would improve the manuscript.

Authors response: Thank you so much for your important comment and suggestions. The overall project interventions were divided into two cohorts run sequentially from 2019-2020 and 2020-2021. While community activities occurred for each cohort, the findings for this study were based on a sample of participants from the first cohort only and participants were engaged at the end of their group activities. Similarly, household members engaged in home visits and partner activities were not repeated across cohorts. Another control measure was to have an external and reputable consulting firm conduct the data collection and analysis for this study. We have added an explanation of these measures to avoid bias and limit spillover in the revised manuscript (page 12). 

♣ Outcome variables – How authors have defined ‘correctly know about modern contraceptive method’? Also, the authors assessed knowledge about the place to obtain a method of FP (yes/no) – it is not clear how they measured it? For example, one participant may know the place to obtain ‘Oral Pills’ and ‘Male Condom’ but not an IUD, so how is this participant categorized as a dichotomous variable?

Authors response: Thank you so much for your important comment. Knowledge of FP was measured by whether respondents had heard about modern contraceptive methods such as Condom (yes/no), Injectables (yes/no), Pill (yes/no), Female sterilization (yes/no), Male sterilization (yes/no), Implants (yes/no), IUCD (yes/no), LAM (yes/no), Emergency contraceptive (yes/no), Standard days methods(yes/no). 

In terms of a place to obtain FP method FP (yes/no), the question (“Do you know of a place where you can obtain a method of family planning”) addressed FP methods in general, not individual methods uniquely. 

♣ Explanatorily variables: Few important explanatory variables such as desire for additional children, sex of living children, women decision-making/autonomy power, currently living with husband or not, and exposure to media are not included in the analysis. Including these important variables in the analysis would improve the paper significantly. Furthermore – it is not clear in the paper how the authors have created a ‘wealth’ variable.

Authors response: Thank you for your insightful comments. Unfortunately, we did not capture information about those explanatory variables so cannot report on any mediating effects. We have mentioned this in our limitations (page 20-21). 

In terms of wealth quintile, we created a wealth score for each individual using principal component analysis (PCA). We ranked each respondent by their assigned scores and divided them into the wealth categories: women with the lowest, middle, and highest socioeconomic status. We have added details in the revised manuscript (page 11-12). 

♣ Data collection: One of the major limitations of this study is the high loss to endline interviews and no mention of attrition analysis. What were the main reasons for the loss to follow up (in endline interviews)? Adding analysis about lost to follow-up participants (from the baseline data) would improve the paper.

Authors response: Thank you so much for your important comment. The attrition seen from baseline to endline is due to adolescent girls’ migration within Nepal for further schooling and some married adolescent girls accompanying their husbands as they migrate to India for seasonal work. Acknowledging the role of migration in our program areas, we have done attrition analysis of participants who were lost to follow up. Finding of attrition analysis has been included in revised manuscript (page 15). We have added some details in the limitation. 

It is also not clear whether or not these were repeated measurements (same respondents interviewed twice??). Was the data collection paper-based? How long was the interviews?

Authors response: Thank you for your questions. These were repeated measurements in which the same respondents were interviewed at baseline and endline. Data collection involved using a digital app, data was collected through Survey Solutions, a software allowing it to be administered using mobile phones, allowing for automated skip patterns, and eliminating the need for data entry from paper surveys. Each interview lasted for 30-45min. We have added details in the revised manuscript (page 12).

♣ Data analysis: Descriptive analysis is used – if this is a repeated measurement why other statistical methods such as the general linear model for repeated measurement was not used? Justification/rationale of selection of variables, and use of higher-level statistical methods to control the effect of explanatory variables on the outcomes variables would really improve the results and conclusions of this manuscript. This may enhance the author’s confidence in reporting the impacts of the intervention on increasing knowledge and use of contraceptives.

Authors response: Thank you so much for this important comment which we expected from the reviewers. Actually, we thought about this and ran the logistic regression model. However, we found no statistically significant result in that model. Thus, we have not included that table in the result section and made our interpretation based on the result of cross tabulation. 

♣ Results: Categorization in ethnicity in Table 1 and Table 3 do not match. Making consistency in these two Tables would improve the paper.

 Authors response: Thank you so much for your suggestion, we have revised it. 

♣ Table 3 – There is a reduction in FP use from baseline to endline among a few groups, for example, in Jajarkot district (38% VS 30%) and among Brahmin/Chhetri ethnicity (35% VS 32%) – I would like to know the likely reason for this? Is it because of data quality?

Authors response: Thank you so much for your comment. We rechecked the data quality and reran our analysis for Jajarkot and Brahmin/Chhetri ethnicity and results are the same. So, the data quality is fine. This is the crude observation that we have reported. The differences may be due to that reason.

♣ Discussion and conclusions: The discussion and conclusion section need to be revised based on how the authors would address my comments on data analysis. Based on the current analysis, I am not fully convinced by the authors’ claim of their intervention has improved knowledge and use of modern contraceptives. The improvement observed between baseline and end-line could be a natural change over time. Therefore, I suggest either improving the data analysis or recognizing these important limitations and revising the discussion and conclusion section of the manuscript accordingly.

Authors response: Thank you so much for your important suggestions. We have revised the discussion and conclusion section considering the limitations. 

Reviewer #2: 

Why is this big nonresponse rate of almost 30%? Please say something on that too!

Authors response: Thank you so much for your important comment. The attrition seen from baseline to endline is due to adolescent girls’ migration within Nepal for further schooling and some married adolescent girls accompanying their husbands as they migrate to India for seasonal work.

Does this mean it was also lower than the national use in 2016 (i.e. 43% in 2016) and why? Your result showing 33% is still lower than the current progress seen in 2016? How does that happen? This will put your findings not to be trustworthy or put its validity at stake?

Authors response: Thank you for your comment. The 43% CPR reported in NDHS 2016 is for married women of reproductive age (15-49 years). However, the CPR reported in this study is for adolescents and young women so, it is low. 

Did you try to evaluate the status of unwanted pregnancies and related morbidities and mortalities in the intervention group? How significant was that when compared with your baseline records? Otherwise concluding with this statement will not be possible. Always recommendations should arise from your direct findings!

Authors response: Thank you so much for your comment. We have not evaluated the status of unwanted pregnancies and other related morbidities and mortalities. We have revised the recommendation as per your suggestions. 

Figure should be put according to the journal’s requirement. It will not be easy to also review the data without this figure too. There is no Figure 1 and 2? They should also be shown!

Authors response: Thank you for your suggestion. For the journal requirement, figures were added to a separate file in the manuscript.

Why is this necessary? Is there any difference in lifestyle, culture, language and similar other variables among these ethnic groups? What influence would it bring on the results you would expect? If no difference, we don’t need it!

Authors response: Thank you so much for your insightful comments. Yes, there are differences among these ethnic groups. There is a variation in access and use of FP services. Some of the past studies that have reported inequalities in the use of FP among different caste groups. 

https://www.ncbi.nlm.nih.gov/pmc/articles/PMC4163397/

https://bmjopen.bmj.com/content/bmjopen/12/3/e054369.full.pdf

https://www.demographic-research.org/volumes/vol25/27/25-27.pdf

Summary of the key finding from the cited paper: 

Compared to Brahmins/Chhetris, Newars were nearly twice as likely (AOR: 1.9; 95% CI: 1.4–2.7) to use a modern method while Muslims and Terai Madhesi other castes were least likely.

(https://www.ncbi.nlm.nih.gov/pmc/articles/PMC4163397/

Janajati women were significantly (AOR=2.08, 95% CI 1.16 to 3.71) more likely to use modern contraception compared with Brahmin/Chhetri women.

https://bmjopen.bmj.com/content/bmjopen/12/3/e054369.full.pdf

Does this mean those who were not in the baseline are included in the end line? How did you accommodate the 1% difference among these two groups? Where do they come from?

Authors response: Thank you for your important comment. These percentage are calculated based on the total sample (N=786, n=440, (56%)) in baseline and (N=565, n=320 (58%)) in endline. It means the dropout rate of Brahmin/Chhetri was low in endline and which was high among other ethnic groups. We have added further details in the revised manuscript.

How significant is this progress?

Authors response: Thank you so much for your comment. It was highly significant (p<0.001). We have added this in the revised manuscript.

Here you are showing the difference is significant too. I don’t think if this is true!! I don’t think the difference is significant here too!

Authors response: Thank you so much for your insightful comment. We have rechecked the raw data and we found that there were some data errors in the table which we have now reviewed and corrected in table 3.

If the number of individuals who know about all family planning methods is 11%, how come those who know about modern planning become 10%. It was the subset of the previous variable and to the minimum should be either equal or greater than those who know all family planning methods. How did you put the questions for these two variables in your data collection tool?

Authors response: Thank you so much for your important comment. We have reviewed and corrected it in the revised manuscript. Please see table 3. 

What is your justification for that? It could be a confounder, otherwise please take this variable out. (Janajati high differences in FP use)

Authors response: Thank you so much for your comments. Our finding is consistent with past study in Nepal which reported that Janajati adolescents and young women have higher use of FP than other caste. Some examples of the past studies.

https://bmjopen.bmj.com/content/bmjopen/12/3/e054369.full.pdf

https://www.nepjol.info/index.php/TUJ/article/view/24704/20817

Summary of the key finding from the cited paper: 

Janajati adolescents’ girls and young women were two times more likely (AOR=2.08, 95% CI 1.16 to 3.71) to use modern contraception compared with Brahmin/Chettri 

How did you justify this reduction even though it may not be statistically significant? Jajarkot?

Authors response: Thank you for your important comment. We have rechecked the data quality. We again ran our analysis for Jajarkot, and results are the same. So, the data quality is fine. This is the crude observation that we have reported. The differences may be due to that reason.

This data tells the intervention didn’t work! What was your expectation in their indicators before you state the intervention then? Pairty?

Authors response: Thank you for your important comments. We have mentioned details about this finding in the discussion section. The young women having two or more children had increased their use of FP methods compared to the women with no children. “This could be because the women with two or more children may have already attained their desired number of children and hence were encouraged to use contraceptives to limit their childbirth after being exposed to the Healthy Transition’s interventions”.

This is still a confounder! Being Dalit will never make one to have a lower wealth status, but this can be measured with a simple wealth index measurement without even considering their ethnicities. Always whenever you consider ethnicity you have to make sure being in a specific ethnic group should have a natural contribution eg. Genetic, etc, for the outcome you are measuring. If phenotypic or artificial modified, however, you should avoid considering ethnicity!

Authors response: Thank you for your comment. Yes, there are differences between these ethnic groups. There is variation in access and use of FP services. Some of the previous studies that have reported inequalities in the use of FP among different caste groups. 

https://www.ncbi.nlm.nih.gov/pmc/articles/PMC4163397/

https://www.demographic-research.org/volumes/vol25/27/25-27.pdf

https://www.nepjol.info/index.php/TUJ/article/view/24704/20817

Summary of the key findings from cited paper: 

Compared to Brahmins/Chhetris, Newars were nearly twice as likely (AOR: 1.9; 95% CI: 1.4–2.7) to use a modern method while Muslims and Terai Madhesi other castes were least likely.

https://www.ncbi.nlm.nih.gov/pmc/articles/PMC4163397/

This outcome however is not measured properly. You only measured their knowledge and use of FP methods. How about its impact on the number of births per family? The parity gaps within births? How about the number of unwanted pregnancies happened before and after intervention? How about the number of abortions because of that? Maternal deaths before and after intervention due to unwanted pregnancies or lack of knowledge of appropriate FP services? The authors should consider all these variables to conclude that the intervention is effective and be recommended at a policy level in the future! Please also use data presentation techniques like graphs showing the significance in the progress between the indicated variables.

Authors response: Thank you so much for your insightful comments. In this study, we have only tried to measure the FP outcome only. We have not measured the number of unwanted pregnancies, abortions, and maternal deaths. In terms of data presentation techniques, we tried to use different graphs but since we had several variables in our data set so couldn’t fit them in the bar graph and pie chart, hence we reported our findings in the table only. 

We would like to thank the editor and both the reviewers for their insightful comments and feedback. Thank you so much for inviting us to revision of this manuscript. 

Dipendra Singh Thakuri, on behalf of all co-authors

---

## [Decision Letter · Decision Letter 1]

24 Jan 2023

PONE-D-22-10311R1Improving family planning uptake among adolescents and young women in Western Nepal: Learning from the implementation of  Healthy Transition ProjectPLOS ONE

Dear Dr. Thakuri,

Thank you for submitting your manuscript to PLOS ONE. After careful consideration, we feel that it has merit but does not fully meet PLOS ONE’s publication criteria as it currently stands. Therefore, we invite you to submit a revised version of the manuscript that addresses the points raised during the review process.

We look forward to receiving your revised manuscript.

Kind regards,

Gbenga Olorunfemi, MBBS,MSC,FMCOG,FWASC

Academic Editor

PLOS ONE

Journal Requirements:

Reviewers' comments:

Reviewer's Responses to Questions

**Comments to the Author**

1. If the authors have adequately addressed your comments raised in a previous round of review and you feel that this manuscript is now acceptable for publication, you may indicate that here to bypass the “Comments to the Author” section, enter your conflict of interest statement in the “Confidential to Editor” section, and submit your "Accept" recommendation.

Reviewer #2: (No Response)

2. Is the manuscript technically sound, and do the data support the conclusions?

Reviewer #2: Yes

3. Has the statistical analysis been performed appropriately and rigorously? 

Reviewer #2: Yes

4. Have the authors made all data underlying the findings in their manuscript fully available?

Reviewer #2: No

5. Is the manuscript presented in an intelligible fashion and written in standard English?

Reviewer #2: Yes

6. Review Comments to the Author

Reviewer #2: The paper addresses all the objectives well. The variation in the number of participants at the baseline and endline survey is not well indicated, what happened to those missing individuals. The inclusion and exclusion criteria is not well addressed too.

7. PLOS authors have the option to publish the peer review history of their article (what does this mean?). If published, this will include your full peer review and any attached files.

Reviewer #2: No

---

## [Author Response · Author response to Decision Letter 1]

14 Feb 2023

Feb 14, 2023

Dear Editors-in-Chief

PLOS ONE 

Thank you very much for your email with the decision dated Jan 24, 2023. We found that reviewers' and editor feedbacks were insightful. We have addressed reviewers’ comments point by point. We have uploaded the revised manuscript showing track changes so that you can see all revisions and modifications we have made. We believe that our revisions will satisfy you and both reviewers. We have included a clean copy and track change copy of the revised manuscript. 

Thank you for considering this manuscript for publication.

Sincerely,

Dipendra Singh Thakuri

Reviewer #2: 

The title should be modified as follows: “Implementation of the Health Transition Project has improved family planning uptake among adolescents and young women in Western Nepal: a pre-and post-intervention study.” 

Authors response: Thank you so much for important suggestions/inputs. We have modified the title incorporating your suggestions. The modified title is “Effect of Healthy Transitions intervention in improving family planning uptake among adolescents and young women in Western Nepal: a pre-and post-intervention study”. 

What happen to the rest of the participants who were involved at the beginning of the intervention. From 786-565 girls, why?

Authors response: Thank you for your insightful comment. Rest of the participants (221 AGYW) who involved in the baseline were lost to follow up. The losses observed between baseline and endline were due to adolescent girls pursuing higher secondary education, some married adolescent girls moved to India with their husbands for seasonal labor work and remaining were not surveyed because of their inconsistent participation in the project. And these losses were random.

Where is the figure?? 

Authors response: Thank you for your query. As per the journal guideline, the figure supposed to be uploaded as separate file, so it is uploaded as separate file and not included in the main manuscript. 

No abbreviation in subtitles or titles.

Authors response: Thank you for important suggestion. We have removed abbreviations from subtitle and titles as suggested. 

Also decode them when you use them the first time.

Authors response: Thank you for your suggestions. We have revised it as suggested. 

Not clear or needs some modification.

Authors response: Thank you so much for your important comment. We have rephrased it as suggested. 

One and half page would be enough

Authors response: Thank you so much for your feedback. 

??? Difference

Authors response: Thank you for your query. The difference is from baseline to endline. 

Why did you have to wait 12 months? Six months would be enough, or it would be nice to see the sixth month record as well as the twelfth month to also show how the trend goes? 

Authors response: Thank you for your important inquiry. The Healthy Transition curriculum consisted of a total of 24 sessions, which were conducted among AGYW on a fortnightly basis, so it took one year to cover the entire package of Healthy Transition, so we had to wait 12 months to complete the full intervention package.

The similarities in the proportions at each category at the baseline and endline survey brings a question of trustworthiness of the data. 

Authors response: Thank you so much for comment. We followed the same groups of participants from baseline to endline, and a similar proportion of each category may have been lost, so there may not have been many differences between baseline and endline in each category.

We would like to thank the editor and both the reviewers for their insightful comments and feedback. Thank you so much for inviting us to revision of this manuscript. 

Dipendra Singh Thakuri, on behalf of all co-authors

---

## [Decision Letter · Decision Letter 2]

20 Mar 2023

PONE-D-22-10311R2Effect of Healthy Transitions intervention in improving family planning uptake among adolescents and young women in Western Nepal: a pre-and post-intervention studyPLOS ONE

Dear Dr. Thakuri,

Thank you for submitting your manuscript to PLOS ONE. After careful consideration, we feel that it has merit but does not fully meet PLOS ONE’s publication criteria as it currently stands. Therefore, we invite you to submit a revised version of the manuscript that addresses the points raised during the review process.

We advise that authors should get an experienced biostatistician to review the data analysis especially Tables 3 and Table 4. For comparison of categorical variable of baseline and endline, a Pearson's Chi-square may not be appropriate because there is a violation of the assumption of "independent" groups. A McNamar test appears to be more appropriate while paired ttest is more appropriate for a continuous variable that is normally distributed.  In Table 3 frequencies were stated without percentages.  In the column for "Diff", some places had frequencies. While others had percentages. It makes the Table quite difficult to understand. Authors should revise this.The calculation of absolute difference and corresponding P-value was not stated in the statistical analysis section. Authors should explain fully what statistical analysis that was done (The statistical steps should be apparent/ more explicit). I reckon that the P-value obtained from a Chi-square does not apply to absolute difference of categorical values and/or frequencies as depicted and interpreted in the result. Authors should get a biostatistician to review the data analysis and result presentation so that valid conclusion can be obtained from the data/result.Table 4. Title. Write "FP" in full.Table 4 also requires overhaul of the statistical analysis. 

We look forward to receiving your revised manuscript.

Kind regards,

Gbenga Olorunfemi, MBBS,MSC,FMCOG,FWASC

Academic Editor

PLOS ONE

Journal Requirements:

Reviewers' comments:

Reviewer's Responses to Questions

**Comments to the Author**

1. If the authors have adequately addressed your comments raised in a previous round of review and you feel that this manuscript is now acceptable for publication, you may indicate that here to bypass the “Comments to the Author” section, enter your conflict of interest statement in the “Confidential to Editor” section, and submit your "Accept" recommendation.

Reviewer #2: All comments have been addressed

2. Is the manuscript technically sound, and do the data support the conclusions?

Reviewer #2: Yes

3. Has the statistical analysis been performed appropriately and rigorously? 

Reviewer #2: Yes

4. Have the authors made all data underlying the findings in their manuscript fully available?

Reviewer #2: Yes

5. Is the manuscript presented in an intelligible fashion and written in standard English?

Reviewer #2: Yes

6. Review Comments to the Author

Reviewer #2: yes all the comments were incorporated according to the reviews made. I expect some formating on the manuscript though. It is my expectation PLOS ONE would let the authors to follow such guidelines.

7. PLOS authors have the option to publish the peer review history of their article (what does this mean?). If published, this will include your full peer review and any attached files.

Reviewer #2: **Yes: **Worku N.

---

## [Author Response · Author response to Decision Letter 2]

6 May 2023

April 3, 2023

Dear Editors-in-Chief

PLOS ONE 

Thank you very much for your email with the decision dated March 21, 2023. We found that editor feedbacks were insightful. We have addressed editor comments point by point. We have uploaded the revised manuscript showing track changes so that you can see all revisions and modifications we have made. We believe that our revisions will satisfy you. We have included a clean copy and track change copy of the revised manuscript. 

Thank you for considering this manuscript for publication.

Sincerely,

Dipendra Singh Thakuri

Editor comments: 

We advise that authors should get an experienced biostatistician to review the data analysis especially Tables 3 and Table 4. For comparison of categorical variable of baseline and endline, a Pearson's Chi-square may not be appropriate because there is a violation of the assumption of "independent" groups. A McNamar test appears to be more appropriate while paired t-test is more appropriate for a continuous variable that is normally distributed. 

Author’s response: Thank you so much for your insightful comment. We have consulted with a biostatistician to review the data analysis. Based on the suggestions, we did a re-analysis of Table 3 and Table 4 using the McNamar test and reported the findings in the revised manuscript. 

In Table 3 frequencies were stated without percentages. In the column for "Diff", some places had frequencies. While others had percentages. It makes the Table quite difficult to understand. Authors should revise this.

Author’s response: Thank you so much for your suggestions. We have reviewed and revised it as per the suggestions. 

The calculation of absolute difference and corresponding P-value was not stated in the statistical analysis section. Authors should explain fully what statistical analysis that was done (The statistical steps should be apparent/ more explicit). I reckon that the P-value obtained from a Chi-square does not apply to absolute difference of categorical values and/or frequencies as depicted and interpreted in the result. Authors should get a biostatistician to review the data analysis and result presentation so that valid conclusion can be obtained from the data/result.

Author’s response: Thank you for your important comments. We have added details about statistical analysis in the statistical analysis section of the revised manuscript. In terms of data analysis, we consulted with the biostatistician and reran the analysis of the data using appropriate statistical test. 

Table 4. Title. Write "FP" in full.

Table 4 also requires overhaul of the statistical analysis. 

Author’s response: Thank you for your suggestions. We have reviewed and revised it as per your suggestions. 

We would like to thank the editor and both the reviewers for their insightful comments and feedback. Thank you so much for inviting us to revision of this manuscript. 

Dipendra Singh Thakuri, on behalf of all co-authors

---

## [Editor Report · Decision Letter 3]

23 May 2023

Effect of Healthy Transitions intervention in improving family planning uptake among adolescents and young women in Western Nepal: a pre-and post-intervention study

PONE-D-22-10311R3

Dear Dr. Thakuri,

We’re pleased to inform you that your manuscript has been judged scientifically suitable for publication and will be formally accepted for publication once it meets all outstanding technical requirements.

Kind regards,

Gbenga Olorunfemi, MBBS,MSC,FMCOG,FWASC

Academic Editor

PLOS ONE
---

## [Editor Report · Acceptance letter]

2 Jun 2023

PONE-D-22-10311R3 

Effect of Healthy Transitions intervention in improving family planning uptake among adolescents and young women in Western Nepal: a pre-and post-intervention study 

Dear Dr. Thakuri:

I'm pleased to inform you that your manuscript has been deemed suitable for publication in PLOS ONE. Congratulations! Your manuscript is now with our production department. 

Kind regards, 

on behalf of

Dr. Gbenga Olorunfemi 

Academic Editor

PLOS ONE